# Immunological Analysis of the Hepatitis B Virus “a” Determinant Displayed on Chimeric Virus-Like Particles of *Macrobrachium rosenbergii* Nodavirus Capsid Protein Produced in *Sf*9 Cells

**DOI:** 10.3390/vaccines8020275

**Published:** 2020-06-04

**Authors:** Nathaniel Nyakaat Ninyio, Kok Lian Ho, Hui Kian Ong, Chean Yeah Yong, Hui Yee Chee, Muhajir Hamid, Wen Siang Tan

**Affiliations:** 1Department of Microbiology, Faculty of Biotechnology and Biomolecular Sciences, Universiti Putra Malaysia, UPM Serdang 43400, Malaysia; nathanielninyio@kasu.edu.ng (N.N.N.); yongcheanyeah@hotmail.com (C.Y.Y.); muhajir@upm.edu.my (M.H.); 2Department of Microbiology, Faculty of Science, Kaduna State University, P.M.B. 2339, Tafawa Balewa Way, Kaduna 800241, Nigeria; 3Department of Pathology, Faculty of Medicine and Health Sciences, Universiti Putra Malaysia, UPM Serdang 43400, Malaysia; klho@upm.edu.my (K.L.H.); onghk1991@gmail.com (H.K.O.); 4Department of Medical Microbiology and Parasitology, Faculty of Medicine and Health Sciences, Universiti Putra Malaysia, UPM Serdang 43400, Malaysia; cheehy@upm.edu.my; 5Institute of Bioscience, Universiti Putra Malaysia, UPM Serdang 43400, Malaysia

**Keywords:** virus-like particles, *Sf*9 cells, hepatitis B, *Macrobrachium rosenbergii* nodavirus, cytotoxic T-lymphocytes, natural killer cells, memory B cells, BALB/c mice

## Abstract

Chimeric virus-like particles (VLPs) have been widely exploited for various purposes including their use as vaccine candidates, particularly due to their ability to induce stronger immune responses than VLPs consisting of single viral proteins. In the present study, VLPs of the *Macrobrachium rosenbergii* nodavirus (*Mr*NV) capsid protein (Nc) displaying the hepatitis B virus “a” determinant (aD) were produced in *Spodoptera frugiperda* (*Sf*9) insect cells. BALB/c mice immunised with the purified chimeric Nc-aD VLPs elicited a sustained titre of anti-aD antibody, which was significantly higher than that elicited by a commercially available hepatitis B vaccine and *Escherichia coli-*produced Nc-aD VLPs. Immunophenotyping showed that the *Sf*9-produced Nc-aD VLPs induced proliferation of cytotoxic T-lymphocytes and NK1.1 natural killer cells. Furthermore, enzyme-linked immunospot (ELISPOT)analysis showed the presence of antibody-secreting memory B cells in the mice splenocytes stimulated with the synthetic aD peptide. The significant humoral, natural killer cell and memory B cell immune responses induced by the *Sf*9-produced Nc-aD VLPs suggest that they present good prospects for use as a hepatitis B vaccine candidate.

## 1. Introduction

Hepatitis B has been declared by the World Health Organization (WHO) as one of the major health concerns of the century. The disease is caused by hepatitis B virus (HBV), which belongs to the family *Hepadnaviridae* [1,2]. About 250 million people, globally, are living carriers of HBV, and yearly recorded deaths due to hepatitis B are estimated about 1 million [3]. However, none of the available treatment regimens against hepatitis B is curative, and concerns have been raised with regards to the efficacies of the available vaccines. Some limitations of the available hepatitis B vaccines include their inability to elicit sustained protective immunity in some individuals [4,5], their ineffectiveness in producing protective immunity in chronically infected subjects [6], induction of a poor immune response in about 10% of vaccinated adults [7], and their inability to confer protective immunity in individuals who are unresponsive to yeast-derived vaccines [8]. Therefore, continuous development of more effective hepatitis B vaccines is essential.

Of relevance to the development of improved hepatitis B vaccines is the understanding of the HBV viral structural proteins. The HBV genome codes for three envelop surface antigens (HBsAg), namely the large-HBsAg (L-HBsAg), middle-HBsAg (M-HBsAg) and small-HBsAg (S-HBsAg) [9]. All these three antigens share the same C-terminal domain known as the *S-*domain, and within the 100-169 amino acid residues of this domain lies a highly immuno-dominant region known as the “a” determinant (aD). This aD is highly conserved among various HBV strains [6,9]. Furthermore, during the viral infection, the aD is the main component of the virus that induces the production of HBV-neutralising antibodies within infected hosts [10]. Therefore, this makes it an important epitope to be exploited in the development of hepatitis B vaccines.

Advancements in vaccine development have highlighted virus-like particles (VLPs) as non-infectious and non-replicative virus-derived entities, which can effectively function as vaccine development platforms. These VLP-derived platforms are usually assembled from a single viral protein [11], or a chimeric fusion of multiple viral subunits consisting of a carrier protein and a displayed epitope [12].

*Macrobrachium rosenbergii* nodavirus (*Mr*NV) capsid protein (Nc) has been used as a carrier in vaccine development because it has been shown to prevent carrier-induced epitopic suppression of displayed epitopes [13]. *Mr*NV is the non-zoonotic aetiological agent of the highly infectious white tail disease in *Macrobrachium rosenbergii* (the giant freshwater prawn). *Mr*NV is a non-enveloped bipartite RNA virus, which consists of RNA1 (3.1 kb) and RNA2 (1.2 kb), encoding for the viral RNA-dependent RNA polymerase (RdRp) and Nc, respectively [14]. When foreign epitopes are fused at the C-terminus of *Mr*NV Nc, the chimeric fusion proteins expressed in *Escherichia coli* (*E. coli*) self-assembled into VLPs [13,15,16]. Hence, this has been exploited for the production of vaccine candidates against hepatitis B, in *E. coli* [16]. However, the *E. coli*-produced *Mr*NV VLPs are less stable, and their yield is lesser than those produced in *Spodoptera frugiperda* (*Sf*9) insect cells [17]. Therefore, in the present study, the aD was fused at the C-terminal end of Nc, and the chimeric VLPs were produced in *Sf*9 cells. The immunological efficacy of *Sf*9-produced Nc-aD VLPs was studied in BALB/c mice. Our results showed that the anti-aD antibody elicited by the *Sf*9-produced Nc-aD VLPs was sustained even at day 182 (including 14 days of acclimatisation) following the first administration. In addition to the humoral immune response, the Nc-aD VLPs induced cytotoxic T lymphocyte (CTL) and natural killer (NK) cell activities in the immunised mice. Also, antibody-secreting cells (ASCs) were detected in mice splenocytes stimulated with a synthetic aD peptide, indicating induction of memory B cell immunity by the Nc-aD VLPs. Overall, these *Sf*9-produced Nc-aD VLPs elicited significant humoral, CTL, NK cell and memory immune responses. Our findings suggest that the *Sf*9-produced Nc-aD VLPs displaying the HBV aD present good prospects to be developed as a hepatitis B vaccine candidate.

## 2. Materials and Methods 

### 2.1. Ethics Statement

Animal studies were conducted with the approval of Universiti Putra Malaysia’s Institutional Animal Care and Use Committee (IACUC approval number R026/2018). All procedures in the animal studies strictly complied with the committee’s guidelines. The BALB/c mice were housed in a light and temperature-regulated animal facility designated as biosafety level 2 (BSL-2).

### 2.2. Preparation of Baculovirus Stock

The first passage (P1) of the baculovirus was prepared as previously described [17] with some modifications. Briefly, 8 × 10^5^
*Sf*9 cells (ATCC^®^ CRL-1711™, ATCC, Manassas, VA, USA) were seeded in each well of a 6-well plate, and transfected with bacmid DNA (1 µg) harbouring the *Nc-aD* gene and a 6 × Histidine-tag (His-tag) coding sequence. The bacmid-transfected *Sf*9 cells were cultured in Sf-900 III medium (2 mL; Life Technologies, Carlsbad, CA, USA), supplemented with 4% foetal bovine serum (FBS), and incubated at 27 °C for 4 days. *Sf*9 cell debris and the culture supernatant containing baculovirus were separated by centrifugation, at 200 × *g* for 5 min. The supernatant containing the baculovirus was collected, and kept as P1 stock at 4 °C.

### 2.3. Expression of Nc-aD VLPs

Productions of Nc-aD in *Sf*9 cells and *E. coli* (Invitrogen, Carlsbad, CA, USA) were performed as previously described [17,18]. For the production of Nc-aD VLPs in *Sf*9 cells, the insect cells were grown in Sf-900 III medium supplemented with 4% FBS (Life Technologies, Carlsbad, CA, USA) to 2 × 10^6^ cells/mL density. At a cell confluency of ~80%, the *Sf*9 cells were infected with 10% of the P1 baculovirus stock and incubated at 27 °C for 5 days until ~90% cell death was observed. The culture supernatant and cell debris were separated by centrifugation at 200 × *g* for 5 min. This was followed by a gentle resuspension of the *Sf*9-cell pellet in ice-cold lysis buffer (15 mL; 0.4% Tween 20, 22.6 mM NaH_2_PO_4_, 77.4 mM Na_2_HPO_4_, 1 mM phenylmethylsulfonyl fluoride; pH 7.4), and sonicated three times at 200 W for 10 s with an interval of 2 min between cycles. The cell lysate was separated by centrifugation at 12,000 × *g* for 5 min at 4 °C.

### 2.4. Purification of Nc-aD VLPs

Nc-aD VLPs in the culture supernatant and lysate of *Sf*9 cells were purified using immobilised metal affinity chromatography (IMAC) as previously described [18] with some modifications. First, the culture supernatant and cell lysate were filtered with a syringe filter (0.45 µm). The filtrates were each loaded into a His-Trap HP column (1 mL, GE Healthcare, Buckinghamshire, United Kingdom), which had been washed with sodium phosphate buffer (10 mL; 22.6 mM NaH_2_PO_4_, 77.4 mM Na_2_HPO_4_; pH 7.4). The bound samples were washed with the binding buffer A (10 mL; 500 mM NaCl, 22.6 mM NaH_2_PO_4_, 10 mM imidazole; pH 7.4), and with binding buffer B (10 mL; 500 mM NaCl, 22.6 mM NaH_2_PO_4_, 50 mM imidazole; pH 7.4). The bound protein was eluted with the elution buffer (5 mL; 500 mM NaCl, 22.6 mM NaH_2_PO_4_, 500 mM imidazole; pH 7.4). Then, the eluted protein was dialysed in sodium phosphate buffer (3 L) for 24 h at 4 °C. The dialysed protein was concentrated using a centrifugal protein concentrator (Vivaspin, Sartorius, Gottingen, Germany) with a molecular weight cut-off of 10 kDa, and protein quantification was performed using the Bradford assay. Purification of Nc-aD VLPs produced in *E. coli* was performed as described [18].

### 2.5. Sodium Dodecyl Sulphate-Polyacrylamide Gel Electrophoresis (SDS-PAGE) and Western Blotting

The Nc-aD VLPs were mixed with 6× SDS-PAGE sample loading buffer [0.2% (w/v) bromophenol blue, 4% (w/v) SDS, 100 mM Tris-HCl, pH 6.8, 20% (v/v) glycerol, 200 mM β-mercaptoethanol] and denatured by heating for 10 min. The sample was then loaded into SDS-polyacrylamide gel [12% (w/v)] and electrophoresed at 16 mA for 80 min. The electrophoresed gel was stained with staining solution [0.1% (w/v) Coomassie brilliant blue R-250, 40% (v/v) methanol, 10% (v/v) acetic acid] for 15 min and destained with destaining solution [30% (v/v) methanol, 10% (v/v) acetic acid] until the protein bands became visible.

Proteins samples on an SDS-PAGE gel were transferred onto a nitrocellulose membrane by a semi-dry transblotter (Bio-Rad, Hercules, CA, USA). The membrane was then blocked with skimmed milk [10% (w/v) Anlene, Auckland, New Zealand] in tris-buffered saline (TBS) (50 mM Tris-HCl, 150 mM NaCl; pH 7.4) at room temperature (RT) for 1 h. The blocked membrane was then washed three times with TBS-tween (TBST) buffer [TBS containing 0.1% (v/v) Tween 20] before the anti-His monoclonal antibody (1:5000 dilution in TBS; Invitrogen, San Diego, CA, USA) or, the anti-HBsAg monoclonal antibody (1:2,500 dilution in TBS; MP Biomedicals, Santa Ana, CA, USA)] was added and incubated overnight at 4 °C. The membrane was again washed three times with TBST buffer and incubated with the diluted anti-mouse antibody (1:5000 dilution in TBS; KPL, Milford, MA, USA), or the anti-guinea pig antibody conjugated to alkaline phosphatase (1:5000 dilution; KPL, Milford, MA, USA) for 1 h at RT. Following another washing step with TBST, the membrane was incubated with 5-bromo-4-chloro-3-indolyl phosphate (BCIP) and nitro blue tetrazolium (NBT), with gentle rocking until protein bands became visible. Colour development was stopped by washing the membrane in water. 

### 2.6. Transmission Electron Microscopy 

The Nc-aD proteins (15 µL; 100 ng/µL) purified from *Sf*9 and *E. coli* cells using IMAC were adsorbed to 200-mesh copper grids for 5 min. The grids were then stained with uranyl acetate solution [2% (w/v); 15 µL] for 5 min. The grids were dried in air, and micrographs were taken with a transmission electron microscope (Hitachi H7700, Hitachi, Tokyo, Japan). 

### 2.7. Immunisation of BALB/c Mice

Five to six weeks old female BALB/c mice were randomly assigned to seven immunisation groups (n = 8) and acclimatised for 2 weeks. The mice were then immunised subcutaneously with the vaccine candidate consisting of the Nc-aD VLPs (100 µL; 0.34 mg/mL) and the adjuvant (100 µL; Imject Alum, Thermo Scientific, USA). Nc-aD VLPs purified from the culture supernatant and lysate of *Sf*9 cells were pooled and used for the immunisation. The first and second boosters were administered on days 35 (5th week) and 56 (8th week), respectively. The positive control groups were immunised with Engerix B (100 µL; GlaxoSmithKline, Rixensart, Belgium) and *E. coli*-produced Nc-aD VLPs (100 µL; 0.34 mg/mL), respectively. The negative control groups were immunised with *Sf*9-produced Nc (100 µL; 0.34 mg/mL) and *E. coli*-produced Nc (100 µL; 0.34 mg/mL), respectively. With the exception of Engerix B, the aforementioned positive and negative controls were all mixed with the alum adjuvant (100 µL) before administration.

Blood (100 µL) was collected from each mouse via submandibular bleeding on days 14, 35, 56, 77, 98, 119, 140, 161 and 182, respectively. On days 14, 35 and 56, blood collection was performed before immunisation. Blood samples were kept at RT for 30 min until clotting was observed, followed by two rounds of centrifugation at 1000 × *g* for 10 min at RT. The sera were collected and stored at −80 °C.

### 2.8. Immunogenicity of Nc-aD VLPs

Enzyme-linked immunosorbent assay (ELISA) was performed using the sera collected from immunised mice on days 14, 35, 56, 77, 98, 119, 140, 161 and 182, respectively. A synthetic aD peptide (0.5 µg; GL Biochem, Shanghai, China) was coated overnight on a 96-well microtiter plate at 4 °C. The wells were then washed three times using TBST, and blocked with milk-diluent (200 µL; KPL, Seracare, Milford, MA, USA) at RT for 1 h. The wells were washed again using TBST, and loaded with the diluted serum samples (1:500 dilution in TBS; 100 µL), and incubated for 1.5 h at RT. Again, the wells were washed three times using TBST, the alkaline phosphatase-conjugated anti-mouse monoclonal antibody (1:5000 dilution in TBS; KPL, Milford, MA, USA; 100 µL) was added to each well, and followed by incubation for 1.5 h at RT. Finally, the wells were washed with TBST, and colour development was performed by the addition of *p*-nitrophenyl phosphate (*p*-NPP; 50 µL per well), and incubation in the dark for 20 min. The absorbance measurement was performed at 405 nm using the ELx800 microtiter plate reader (BioTek, Winooski, VT, USA).

### 2.9. Immunophenotyping of Mouse Splenocytes

BALB/c mice (n = 4) from each immunisation group were sacrificed on day 63 (7 days after the administration of the second booster), their spleens were harvested, and kept in ice-cold PBS (pH 7.4). The spleens were homogenized on a cell strainer (70 µm) in the presence of ice-cold PBS (1 mL). The homogenate was separated by centrifugation at 2000 × *g* for 5 min at 4 °C, and the pellet of splenocytes was resuspended gently in ice-cold erythrocyte lysis buffer (155 mM NH_4_Cl, 0.1 mM EDTA and 10 mM KHCO_3_; pH 7.4, 5 mL), followed by incubation on ice for 10 min. Then, ice-cold PBS (6 mL) was added, and the splenocytes suspension was centrifuged at 2000 × *g* for 5 min at 4 °C. Erythrocyte lysis was repeated until the pellet of splenocytes obtained after centrifugation was devoid of red blood cells.

The cell pellet was gently resuspended in ice-cold PBS (1 mL) containing BSA (1%), and aliquots of 5 × 10^6^ cells were dispensed into 1.5 mL tubes. In the dark, anti-CD3 monoclonal antibody, fluorescein isothiocyanate (FITC; Thermo Scientific, Waltham, MA, USA; 0.5 µL), anti-CD4 monoclonal antibody, phycoerythrin (PE; Thermo Scientific, Waltham, MA, USA; 1.25 µL), anti-CD8 monoclonal antibody, allophycocyanin (APC; Thermo Scientific, Waltham, MA, USA; 1.25 µL) and anti-NK1.1 monoclonal antibody (FITC; Thermo Scientific, Waltham, MA, USA; 1 µL) were added to the splenocytes, and incubated on ice for 1 h. The antibody-stained splenocytes were washed with PBS (1 mL) containing BSA (1%), at 2000 × *g* for 5 min at 4 °C. Finally, fixation was performed with paraformaldehyde (1% in 1 mL PBS) and the splenocytes were kept on ice for 24 h. Analysis of the splenocytes (1 × 10^5^ splenocytes) was performed using a flow cytometer (NovoCyte; Acea Biosciences, San Diego, CA, USA). Flow cytometry data were analysed using the NovoExpress software (version 1.3.0, Acea Biosciences, San Diego, CA, USA).

### 2.10. Memory B-cell ELISPOT Assay

ELISPOT assay was performed using the Mouse IgG Single-Colour ELISPOT kit (CTL, Bon, Germany), according to the manufacturer’s protocol with some modifications. Briefly, the spleens of mice (n = 4) from each immunisation group were harvested on day 182. Then, 1 × 10^6^ splenocytes were treated with the polyclonal activator (1:1000 dilution in CTL medium) and incubated for 4 days at 37 °C with 10% CO_2_. Then the splenocytes (1.25 × 10^5^) treated with the polyclonal activator were added into the wells of a 96-well ELISPOT plate, which had been coated with a synthetic aD peptide (10 µg). The splenocytes were incubated at 37 °C for 24 h in 10% CO_2_. The plates were then washed two times with PBS (pH 7.4) and two times with PBST (PBS containing 0.05% Tween-20). Then, the biotin-labelled anti-murine IgG antibody (80 µL, 1:400 dilution in PBS) solution was added per well, followed by overnight incubation in the dark at 4 °C. The wells were again washed three times with PBST, and alkaline phosphatase-conjugated streptavidin solution (80 µL; 1:1000 dilution in PBS) was added. Plate incubation was performed for 1 h at RT, away from light. The wells were again, washed twice with PBST and twice with distilled water. Colour development was performed by the addition of the blue developer solution (80 µL), and the plate was incubated for 20 min at RT, away from light. The reaction was stopped by gently rinsing the plate three times with tap water, and the plate was air-dried. Finally, the spots formed by antibody secreting cells were counted with the ELISPOT plate reader (CTL Immunospot S6 Ultimate, CTL, Cleveland, OH, USA). The results were considered as positive based on two criteria: if ASCs were observed in at least 2 out of the triplicate measurements, and if the number of observed ASCs was at least twice the number of ASCs in the negative controls.

### 2.11. Statistical Analysis

Variations in the immunogenicity of the various administered injections tested using ELISA, variations in the number of gated mice splenocytes during immunophenotyping, and variations in the number of spots counted in ELISPOT, were analysed with the one-way analysis of variance (ANOVA). The Duncan’s multiple-range tests were used. *p* values less than 0.001 are considered as significant, and those less than 0.0001 are considered as extremely significant. The statistical analysis was performed using the IBM SPSS statistics software (version 23, IBM, Armonk, NY, USA).

## 3. Results

### 3.1. SDS-PAGE and Western Blotting of Purified Nc-aD VLPs Produced in Sf9 Cells

Nc-aD VLPs were purified by IMAC from the culture supernatant and cell lysate of the *Sf*9 cells infected with 10% of the P1 baculovirus stock. The purified Nc-aD was eluted from the IMAC column using 500 mM imidazole, dialysed in sodium phosphate buffer, and analysed using SDS-PAGE and western blotting. A protein band of ~52 kDa was detected (Figure 1), which corresponds to the calculated molecular mass of Nc-aD (51.3 kDa). Transmission electron microscopic analysis revealed that the Nc-aD produced in *Sf*9 assembled into spherical VLPs of different sizes, ranging from ~21 nm to ~55 nm, while the *E. coli*-produced Nc-aD protein assembled into VLPs of ~30 nm in diameter (Figure 2).

### 3.2. Immunogenicity of the Nc-aD VLPs in BALB/c Mice

The Nc-aD VLPs were subcutaneously administered into BALB/c mice, and the collected serum samples were analysed with ELISA. As shown in Figure 3, there was a significant increase in the anti-aD antibody titre of the immunised mice after the administration of the primary dose of *Sf*9-produced Nc-aD. The administration of the first and second boosters resulted in a significant increase in antibody titre (*p* < 0.001). The highest antibody titre was recorded from serum samples collected on day 77, which corresponds to 21 days after the administration of the second booster. A decline in the antibody titre was observed in sera from days 98 and 119, after which the antibody titre appeared to reach a plateau. In general, the antibody titre elicited by *Sf*9-produced Nc-aD VLPs was higher than those of *E. coli*-produced Nc-aD VLPs and Engerix B.

### 3.3. Immunophenotyping of Mouse Splenocytes

Immunophenotyping was performed on day 63 after the administration of the second booster in order to quantitate the helper T lymphocytes (CD3^+^ and CD4^+^ cells), CTL (CD3^+^ and CD8^+^ cells) and NK cells (NK1.1^+^ cells) in the splenocytes of sacrificed mice (Table 1). In comparison to the immunisation group injected with buffer only, the groups immunised with *Sf*9-produced Nc-aD VLPs, *E. coli*-produced Nc-aD VLPs and Engerix B had a significantly higher CD8^+^/CD4^+^ ratio (*p* < 0.001), which is an indication of increased CTL activities. 

Figure 4 shows the percentage distribution of NK1.1+ cells per 1 × 10^5^ analysed splenocytes. Overall, the Nc-aD VLPs produced in *Sf*9 had a significantly higher induction of NK cells (*p* < 0.001). Interestingly, *Sf*9-produced Nc VLPs, without the aD, induced a significantly higher NK cell response than *E. coli*-produced Nc-aD VLPs and Engerix B. Also, there was no significant difference between the percentage of NK1.1+ cells in mice inoculated with *E. coli*-produced Nc-aD VLPs and *E. coli*-produced Nc VLPs. These observations suggest that the Nc VLPs alone have some effects on NK cell induction.

### 3.4. ELISPOT of Memory B Cells

On day 182, the mice from all immunisation groups were sacrificed and their splenocytes were stimulated with a synthetic aD peptide to assay the presence of memory B cells or ASCs. The ASCs were counted as spots on the ELISPOT plate following colour development (Table 2). The results revealed that the mice immunised with *Sf*9-produced Nc-aD VLPs, *E. coli*-produced Nc-aD VLPs and Engerix B had significant higher numbers of ASCs per 1.25 × 10^5^ splenocytes (*p* < 0.001), as compared to the groups injected with buffer and Nc VLPs produced in *Sf*9 and *E. coli*. Interestingly, splenocytes harvested from the mice immunised with *Sf*9-produced Nc-aD VLPs had the highest memory B cell response.

## 4. Discussion

In the absence of any effective treatment for hepatitis B infection, the administration of vaccines remains the most effective control measure [19]. However, most of the currently available hepatitis B vaccines have been reported to have some limitations with regards to their ability to effectively protect vaccinated subjects [6,7,8,20]. Therefore, continuous development of better vaccine candidates is necessary. VLPs have been widely exploited as a vaccine development platform, and the choice of an expression system may determine the yield, quality and immunogenicity of the vaccine candidates. A few approaches have been employed for the production of VLPs intended for use as improved hepatitis B vaccine candidates. Unlike the eukaryotic expression system, VLPs produced in bacteria are not post-translationally modified, therefore the former has been widely used for developing HBV vaccine candidates. Post-translational modifications, particularly *O-* and *N-*glycosylation, can enhance the immunogenicity and biological activity of recombinant VLPs. Hyakumura et al. [21] demonstrated that the introduction of extra *N-*glycosylation sites in the S-HBsAg yielded more densely glycosylated VLPs with enhanced immunogenicity, when the particles were expressed in HEK293T mammalian cells. They demonstrated a significant correlation between glycan abundance and immunogenicity of S-HBsAg in BALB/c mice. However, VLP production in mammalian cell lines is often hampered by high production cost and low VLP yield, as compared to yeast cells. These single-celled fungi also have some drawbacks as yeast-produced VLPs are non-secretory, which complicates the downstream purification process due to the presence of cellular debris. HBsAg expressed in *Pichia pastoris* was shown to be localised within the endoplasmic reticulum, and extraction from detergent-treated cell lysates was only possible after several downstream steps [22]. The study also showed that the HBsAg did not assemble into VLPs within the yeast cells, and the particles are most likely formed during the downstream purification process. Purification of HBsAg VLPs produced by *Saccharomyces cerevisiae* involved laborious steps, which include the lysis of cells with detergent and alumina, followed by lyophilisation of the crude lysate, and separation of the particles using sepharyl-S column chromatography and size-exclusion chromatography [23]. Since the 1990s, expression of HBsAg VLPs has been performed in plants, particularly the tobacco [24]. However, concerns have been raised regarding the likelihood of contaminations by toxic plant products and allergens, as well as the non-uniform localisation of the VLPs in plant tissues [25]. HBsAg expression in insect cells has been successfully performed using the baculovirus expression system in the *Sf*9 cell line [26]. The baculoviral expression system is beneficial for its high VLP yield, cost effectiveness and its ability to produce *N-*glycosylated VLPs [27]. Although baculoviruses are incapable of infecting mammalian cells and replicating within them [28], concerns over the possibility of baculoviral contamination of insect cell-produced VLPs intended for use as vaccines have been raised. In addition, cell surface display has been employed for displaying the aD in the production of hepatitis B vaccine candidates. Kim and Yoo [29] displayed the aD on the surface of *E. coli* by fusing the epitope to the ice nucleation protein, an outer membrane protein of *Pseudomonas syringae*.

With regards to the *Mr*NV VLPs, previous studies by our group suggested that the *Sf*9 expression system is a more effective system than *E. coli*, for the production of VLPs for immunologic and structural studies, owing to the fact that the VLPs produced in *Sf*9 are more stable than those produced in *E. coli* [17,30]. The *Sf*9-produced Nc VLPs were stable from pH 2 to 12, and from 5 to 45 °C [17]. In addition, the diameter of the *Sf*9-produced Nc VLPs is 10 nm larger than those produced in *E. coli* [17]. Fifis et al. [31] demonstrated that the size of VLPs plays an important role in inducing immune responses in mice.

The Nc-aD VLPs produced in *Sf*9 cells using the baculovirus expression system were purified using IMAC. The VLPs were heterogeneous in size, ranging from ~21 nm to ~55 nm in diameter, as observed under a transmission electron microscope. The heterogeneous population of the VLPs could be attributed to the fusion of aD at the C-terminal end of Nc. Kueh et al. [17] demonstrated that the Nc produced in *Sf*9 cells assembled into homogenous VLPs with a diameter of ~40 nm. Fusion of a small ubiquitin-like modifier (SUMO) protein at the N- or C-terminus of the Nc of a closely related shrimp nodavirus, *Penaeus vannamei* nodavirus (PvNV), resulted in heterogeneous VLPs [32]. The additional fusion protein is believed to increase the freedom of dimeric capsomeres, which could result in the formation of heterogeneous VLPs [32]. Katsarou et al. [33] showed that the *Sf*9-produced enhanced green fluorescent protein (eGFP) fused to the C-terminus of the core protein of hepatitis C virus assembled into heterogeneous VLPs ranging from 30 to 55 nm in diameter. Another study by Luque et al. [34] revealed that the capsid protein of rabbit haemorrhagic disease virus with N-terminally displayed T cell epitope from chicken ovalbumin and B cell epitope from feline calicivirus capsid protein, assembled into heterogeneous VLPs ranging from 40 to 50 nm in diameter, when the chimeric proteins were expressed in *Sf*9 cells.

The Nc-aD VLPs purified from the cell lysate and supernatant were pooled before being administered subcutaneously into the mice. Administration of three doses of *Sf*9-expressed Nc-aD VLPs into the BALB/c mice elicited a significant production of aD-specific antibody, which was maintained to the endpoint of the study (day 182), demonstrating that the high antibody titre was sustained for 126 days after the second booster injection. The Antibody level reached a plateau from days 119 to 182, suggesting a very slow rate of decline in the elicited humoral immune response (Figure 3). This is in good agreement with a study in which BALB/c mice immunised with the potato-derived hepatitis B surface antigen (HBsAg) elicited anti-HBsAg IgG titre that sustained for ~7 months [11]. It is believed that the sustained antibody production is important in conferring protection on immunised subjects against HBV [35]. In a previous report, HBV-infected bone marrow-transplant recipients experienced HBV-viral clearance [36]. This viral clearance was believed to have been induced by the humoral immunity of the vaccinated bone marrow donors [36].

When compared to the *E. coli*-produced Nc-aD VLPs and the commercial hepatitis B vaccine, Engerix B, the antibody titre elicited by the *Sf*9-produced Nc-aD VLPs was significantly higher at all points studied. Interestingly, the lowest antibody titre recorded in mice immunised with the *Sf*9-produced Nc-aD VLPs (at day 182) was significantly higher than the antibody titres of all other groups even at their peaks (day 77). Ryu et al. [37] demonstrated that the antibodies induced by aD were 2000-fold more specific in viral neutralisation as compared to those induced by the whole HBV viral particle. Hence, the ability of the *Sf*9-produced Nc-aD VLPs to elicit such a significant titre of anti-aD antibody presents good prospects as a new hepatitis B vaccine candidate.

Administration of *Sf*9-produced Nc-aD VLPs into the mice induced CTL proliferation, as evidenced by a higher CD8+/CD4+ ratio in this immunisation group, but not in the negative control groups (buffer only, buffer and alum, *Sf*9-produced Nc VLPs, and *E. coli*-produced Nc VLPs). Although the increase in CTL ratio induced by *E. coli*-produced Nc-aD VLPs (0.69) and Engerix B (0.66) is higher than that induced by *Sf*9-produced Nc-aD VLPs (0.65), the difference is not statistically significant. The observed CTL proliferation could have been induced by the aD component of the immunogens. This is corroborated by a study by Pride et al. [38] who demonstrated that a 15-amino acid peptide homologous to the “a” determinant significantly induced the production of T-lymphocytes and HBsAg-specific antibodies in mice. During HBV infection, CTLs disrupt the expression of HBV genes within hepatocytes by the secretion of interferon-gamma (IFN-γ) and tumour necrosis factor-alpha (TNF-α). It has also been shown that this CTL-mediated disruption of HBV replication is accompanied by the destruction of only a few hepatocytes [39,40]. The proliferation of CTLs during HBV infection has been directly linked to viral clearance, and conversely, suppression of CTL function increased HBV replication in mice [41,42]. It is believed that the induction of CTL proliferation removes HBV infected hepatocytes [43]. A study addressing the protective immunogenicity of a recombinant HBsAg-HBcAg vaccine in C57BL/6 HBV1.3.32 transgenic mice was performed [44]. The mice harbouring the HBV genome showed elevated CTL responses, total seroconversion and a significant decrease in circulating HBV titres after the second injection. Since the aD is the immunodominant region of the HBsAg, the aforementioned study suggests that the Nc-aD-induced humoral and CTL immunity could lower HBV titres *in vivo.*

Here, we also demonstrated that the administration of *Sf*9-produced Nc-aD VLPs resulted in an increase in the NK cell population in the immunised mice (Figure 4), of which those inoculated with *Sf*9-produced Nc-aD VLPs showed the highest NK cell induction, followed by *E. coli*-produced Nc-aD VLPs. The results also showed that *Sf*9-produced Nc VLPs and *E. coli*-produced Nc VLPs significantly induced NK cells. This suggests that the Nc VLPs have some adjuvanting properties, which resulted in the expansion of the NK cell population. The self-adjuvanting properties of VLPs are generally attributed to their size which facilitates the size-dependent uptake by dendritic cells [31]. In addition to inorganic substances that have immunogenicity-enhancing effects, VLPs employed as display platforms for antigenic epitopes have also been recently used as adjuvants, provided that they are able to interact with the surface molecules of dendritic cells [45,46]. The induction of NK cells is important as it is a key component of the innate immunity, which confers protection during viral infections before the adaptive immunity is developed. This suggests that the immunogen, aD, induces the innate immunity of the mice, which is important especially with regards to the role of NK cells in CTL activation, macrophages stimulation, and the direct destruction of virally infected cells. In a previous study that involved chimpanzees infected acutely with HBV, it was shown that the NK cell-activities alone led to a significant decline in HBV DNA even before the T-cell phase of viral clearance was initiated [47].

Although we observed a sustained antibody response in the immunised mice, a decline in antibody titre over time remains a possibility. Therefore, to investigate the presence of memory B cell responses, a memory ELISPOT assay was conducted on mice splenocytes. ELISPOT assay of the splenic memory B cell using the mice splenocytes (Table 2) showed that *Sf*9-produced Nc-aD VLPs induced the production of antigen-specific memory B cells when the splenocytes were treated with a polyclonal activator on day 182. ASCs were detected in splenocytes of mice inoculated with *Sf*9-produced Nc-aD VLPs, *E. coli*-produced Nc-aD VLPs and Engerix B. The results showed that the mice immunised with *Sf*9-produced Nc-aD VLPs had a splenic memory B cell count that was four-fold higher than those immunised with *E. coli*-produced Nc-aD VLPs, and two-fold higher than the mice immunised with Engerix B. To assay the longevity of the immunogenicity of administered hepatitis B vaccines, similar studies were carried out to detect the presence of HBsAg-specific memory B cells in human subjects. In the reported cases, however, the ELISPOT assay detected the presence of the memory B cells in peripheral blood mononuclear cells isolated from the blood of the subjects [48,49]. Although these findings cannot be extrapolated to humans, we believe that the higher number of memory B cells induced by *Sf*9-produced Nc-aD VLPs indicates its ability to confer a long term prophylaxis to immunised candidates against further HBV infections even when antibody titres decline.

## 5. Conclusions

Nc-aD VLPs were produced in *Sf*9 cells and purified by IMAC. Three doses of the *Sf*9-produced Nc-aD VLPs (100 µL; 0.34 mg/mL) were administered, at 21 days interval, into BALB/c mice. The chimeric VLPs induced a robust antigen-specific humoral immune response, which was sustained to 126 days after the second booster dose, until the end-point of the animal study. Immunophenotyping revealed that the *Sf*9-produced Nc-aD VLPs induced CTL and NK cell activities in the immunised mice. ELISPOT assay for memory B cells showed that the Nc-aD VLPs induced the production of ASCs detected in polyclonally activated mice splenocytes stimulated with the synthetic aD peptide on day 182, indicating the induction of a memory B cell immunity. By and large, the *Sf*9-produced Nc-aD VLPs induced significant humoral, CTL, NK cell and memory B cell immune responses.

## Figures and Tables

**Figure 1 vaccines-08-00275-f001:**
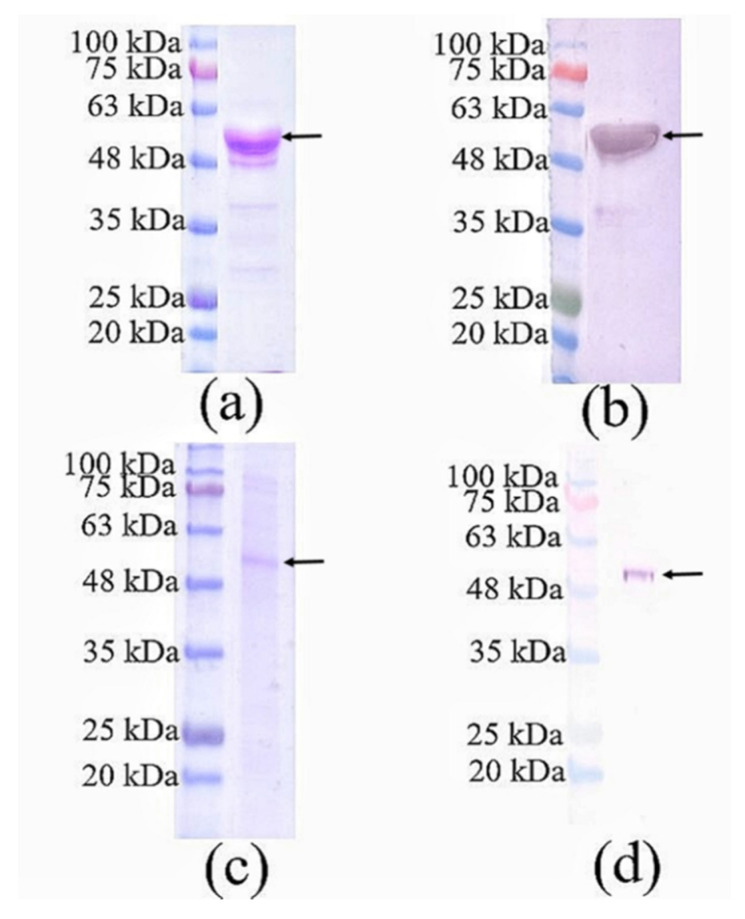
SDS-PAGE and western blotting analysis of Nc-aD produced in *Sf*9 cells and purified using immobilised metal affinity chromatography. (**a**) SDS-PAGE and (**b**) western blotting of Nc-aD purified from the culture supernatant of the *Sf*9 cells infected with 10% of the baculovirus stock. (**c**) SDS-PAGE and (**d**) western blotting of Nc-aD purified from the cell pellet of the *Sf*9 cells infected with 10% of the baculovirus stock. A protein band of ~52 kDa in size was detected as indicated by the arrows. Western blotting was performed with the anti-His monoclonal antibody.

**Figure 2 vaccines-08-00275-f002:**
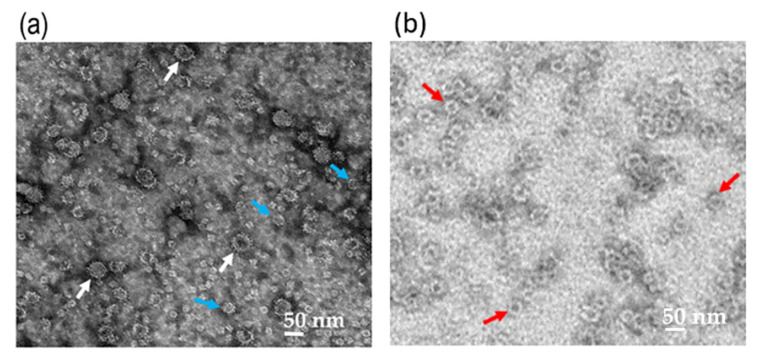
Transmission electron micrograph of the Nc-aD protein. Transmission electron micrographs of the *Sf*9-produced Nc-aD protein (**a**), and *E. coli*-produced Nc-aD protein (**b**). Transmission electron microscopic analysis showed that the *Sf*9-produced Nc-aD protein assembled into virus-like particles (VLPs) ranging from ~21 nm (blue arrows) to ~55 nm (white arrows) in diameter, while the *E. coli*-produced Nc-aD protein assembled into VLPs of ~30 nm (red arrows) in diameter. The white bars indicate 50 nm.

**Figure 3 vaccines-08-00275-f003:**
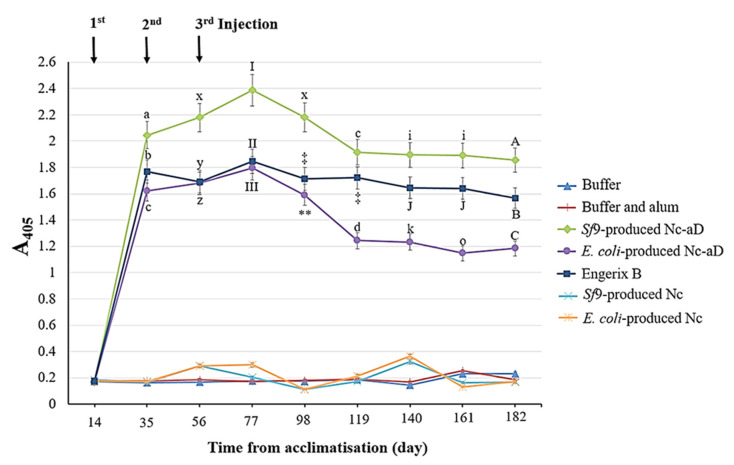
Immunogenicity of the *Sf*9-produced Nc-aD VLPs in BALB/c mice. ELISA was performed to determine the immunogenicity of the *Sf*9-produced Nc-aD VLPs and the control groups. The ELISA plate was coated with a synthetic aD peptide, and reacted with serum samples from the immunised mice. Mice were acclimatised from day 0 to 14 and immunised with the primary injection on day 14. The first and second booster doses were administered on days 35 and 56, respectively. The letters (a, b, c, d, A, B, C, i, J, k, o, x, y, z) and symbols (I, II, III, ‡, ** ) represent the statistical significance of the measurements at each time point. Measurements marked by different letters/symbols are significantly different (*p* < 0.001), and those marked by the same letters/symbols are not significantly different. The error bars indicate the standard deviation (SD) of triplicate measurements.

**Figure 4 vaccines-08-00275-f004:**
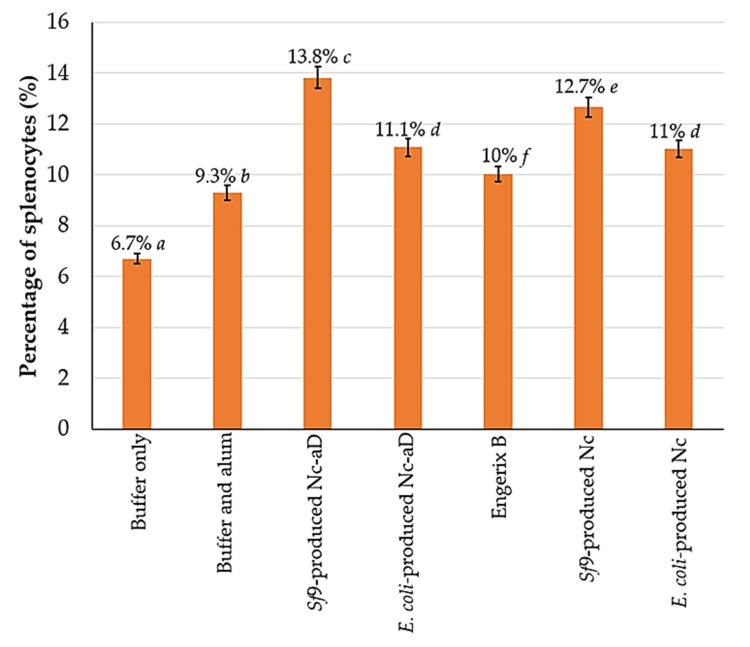
Frequency of NK1.1^+^ mouse splenocytes. Splenocytes from mice sacrificed on day 63 were probed with the anti-NK1.1 monoclonal antibody labelled with FITC. The percentage values shown above represent the percentage of NK1.1^+^ splenocytes from the 1 × 10^5^ probed cells analysed with a flow cytometer. The error bars indicate the standard deviation (SD) of triplicate measurements. The letters (a, b, c, d, e, f) shown above the bars represent the statistical significance of the measurements. Percentage values which are marked by different letters are significantly different (*p* < 0.001), and those marked by the same letters are not significantly different.

**Table 1 vaccines-08-00275-t001:** T lymphocyte populations in mouse splenocytes.

Immunisation Groups	Frequency of Gated Splenocytes (%)		
	CD3^+^ CD4^+^	CD3^+^ CD8^+^	CD8^+^/CD4^+^ Ratio
Buffer only	21.62 ± 0.95 ^a^	11.42 ± 0.51 ^g^	0.53 ± 0.01 ^v^
Buffer and alum	19.70 ± 0.30 ^b^	11.51 ± 0.02 ^g^	0.58 ± 0.01 ^v^
*Sf*9-produced Nc-aD VLPs	13.95 ± 0.15 ^c^	9.07 ± 0.23 ^h^	0.65 ± 0.02 ^w^
*E. coli*-produced Nc-aD VLP_S_	11.81 ± 0.32 ^d^	8.17 ± 0.08 ^i^	0.69 ± 0.01 ^w^
Engerix B	17.95 + 0.42 ^e^	11.92 + 0.12 ^k^	0.66 + 0.02 ^w^
*Sf*9-produced Nc VLPs	13.48 ± 0.45 ^c^	7.63 ± 0.09 ^j^	0.57 ± 0.02^v^
*E. coli*-produced Nc VLPs	17.63 ± 0.37 ^e^	9.69 ± 0.36 ^h^	0.54 ± 0.03 ^v^

The letters (a, b, c, d, e, g, h, i, j, k, v, w) represent the statistical significance of the measurements. T lymphocyte populations which are marked by different letters are significantly different (*p* < 0.001), and those marked by the same letters are not significantly different.

**Table 2 vaccines-08-00275-t002:** Enzyme-linked immunospot (ELISPOT) analysis of activated mice splenocytes.

Immunisation Groups	Spot Count ^1^
Buffer only	0.67 ± 0.58 ^a^
Buffer and alum	0.00 ± 0.00 ^a^
*Sf9*-produced Nc-aD VLPs	26.67 ± 0.57 ^b^
*E. coli*-produced Nc-aD VLPs	6.50 ± 0.45 ^c^
Engerix B	11.30 + 0.71 ^d^
*Sf*9-produced Nc VLPs	0.33 ± 0.58 ^a^
*E. coli*-produced Nc VLPs	0.33 ± 0.58 ^a^

^1^ Spot counts represent ASCs per 1.25 × 10^5^ splenocytes. The letters (a, b, c, d) represent the statistical significance of the measurements. Spot count values which are marked by different letters are significantly different (*p* < 0.001), and those marked by the same letters are not significantly different.

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
