# Peer review of "Immunological Analysis of the Hepatitis B Virus “a” Determinant Displayed on Chimeric Virus-Like Particles of Macrobrachium rosenbergii Nodavirus Capsid Protein Produced in Sf9 Cells"

_vaccines, 2020, doi:10.3390/vaccines8020275_

Round 1

Reviewer 1 Report

Manuscript Title: Immunological Analysis of the Hepatitis B Virus ‘a’ Determinant Displayed on Chimeric Virus-like Particles of Macrobrachium rosenbergii Nodavirus Capsid Protein Produced in Sf9 Cells

General comments:

In this study the authors have studies a new chimeric virus-like particle (VLP) against hepatitis B virus. The authors have used Macrobrachium rosenbergii nodavirus (MrNV) capsid protein (NC) displaying the hepatitis B virus ‘a’ determinant (aD) protein produced in Sf9 cells. The authors found that the Sf9 produced VLPs are more antigenic when compared to the similar VLPs produced in E.coli and also when compared to commercially available vaccine EngerixB. This is a very interesting study with significant implications in developing better vaccines against Hepatitis B. I have few major concerns, which once addressed; the manuscript should be accepted for publication.

Comments:

  • What is the reason of such heterogeneous size differences of VLPs observed in the TEM? Please explain it in the text.
  • Please provide TEM images of coli produced VLPs for side-by-side comparisons with the Sf9 VLPs. Also better TEM images should be provided.
  • Which purified protein was used to measure all the immunological activity: VLPs isolated from the supernatant or the lysate? Was there any difference between the two fractions both in the size heterogeneity and function?
  • In all the figures and tables please cluster the data for Sf9 produced Nc-aD VLPs, coli produced NC-aD VLPs and Engerix B together so that the differences can be more apparent.
  • The authors have not provided any information regarding the stability of the Sf9 produced NC-aD VLPs. The authors should compare side by side various parameters for both Sf9 and coli NC-aD VLPs like pH stability and thermostability.
  • Remove Fig.4 from the article, and its description from the text, as it does not show any meaningful information as the NK cell induction is even seen in NC-VLPs alone in significant amount. In fact there is no difference between the coli produced NC-aDVLPs and the NC-VLPs but the authors still claim in the discussion that it provides better NK cell activation than the commercial vaccine which is misleading.
  • The whole discussion section needs to be re-written, as it seems that the authors have just described the results again here in this section. The authors should discuss more about how their results co-relate with what is known in the field for both Hepatitis B and other viruses where similar or different approaches have been taken for preparing better vaccine candidates.

Author Response

Please see to the attachment.

Reviewer 2 Report

This manuscript presented a study involving immunological studies per a VLP-based vaccine update using insect cell-producing system. The study includes vaccine VLP construction, production and vaccination in mouse models. All experiments formed into a cohesive immunological research to evaluate vaccination effect per adaptive responses at T-, B- and NK-cell levels, and postulated a better vaccine candidate than current VLP vaccine produced in a bacterial system. Overall, it is a nice study and presentation and appropriate to publish in Vaccine Journal. Two points need the author to address during the revision:

1) No viral challenge and infection protective results? Will be better if possible to add or discuss at least      

2) The rationale to include two boosters, and lack of justification and comparison, eg. using just one booster.  

3) There are some typos and insistent reference formatting need to be corrected.

Round 2

Reviewer 1 Report

I thank the authors for their point by point response. This resubmitted article has improved a lot from the previous one and answers most of my concerns. I therefore would like this article to get published in this journal.